# Achieving Quality and Effectiveness in Dementia Using Crisis Teams (AQUEDUCT): a randomised controlled trial evaluating the impact of a best practice Resource Kit used by teams managing crisis in dementia

People with dementia frequently experience mental health crisis requiring psychiatric hospital admission. In the UK, Teams Managing Crisis in Dementia (TMCDs) vary in structure and practice due to the absence of a standardized model. A pragmatic, randomised controlled trial (RCT) was designed to evaluate the AQUEDUCT Best Practice Tool and online Resource Kit (RK). Twenty-three TMCDs across England were randomised 1:1 To receive the RK plus usual care (intervention) or usual care alone (control) (www.isrctn.com/ISRCTN42855694). The primary outcome was the number of psychiatric hospital admissions for people with dementia at the primary endpoint of six months. Secondary outcomes included TMCD staff mental health (GHQ-12), psychological flexibility (WAAQ), and work engagement (UWES); and for people with dementia and carers, service satisfaction (CSQ-8) and mental wellbeing (GHQ-12). There was no significant difference in number of psychiatric admissions between groups (incident rate ratio: 0.74; 95% CI: 0.37-1.48; $p = 0.397$) and the primary endpoint was met. No significant differences were found for the secondary outcomes across staff or service user groups. Fidelity to the intervention varied; five TMCDs met or exceeded implementation criteria, while others reported structural barriers. Limited engagement was attributed to the absence of a learning collaborative and pandemic-related service pressures. Although the RK was valued by staff for guiding quality improvement, it did not significantly reduce hospital admissions or improve secondary outcomes. Future studies should prioritise implementation support and explore systemic barriers to service improvement in dementia crisis care.

There are an estimated 944,000 people living with dementia in the UK[1], and this rising prevalence poses great challenges for healthcare systems worldwide[1,2]. The best practice in dementia care is to support people at home where possible[3], but many people with dementia experience crises[4], which can lead to hospital admissions[5–7]. In the UK from 2012/2013 to 2017/2018, there was a 35% increase in emergency admissions for people with dementia, with costs of £280 million to the NHS[8]. Preventing hospital admissions and decline in independence

✉ e-mail: M.Orrell@nottingham.ac.uk

could also reduce the risk of hospital-related complications, reduce the distress and extra challenges for people with dementia and their carers, and reduce health service costs[9].

In the UK, services to support people with dementia experiencing crisis aim to prevent unnecessary inpatient admissions and provide short-term, high-intensity support at home during times of crisis and when independence is compromised. These services can help resolve crises and reduce admissions[10]. They have a variety of names, including Home Treatment Teams or Crisis Services, which this paper describes as teams managing crisis in dementia (TMCD)[6,7,11,12]. This rapid response and effective management can be essential to avoid mental health admissions, which incur major costs to health services, often increase carer stress, and can be very detrimental to individuals with dementia[6,7,11–13]. Arguably, the ongoing impact of COVID-19 has compounded this further[13].

Achieving Quality and Effectiveness in Dementia using Crisis Teams (AQUEDUCT) is a research programme funded by the National Institute for Health Research (NIHR RP-PG-0612-20004) designed to develop and evaluate a newly developed model of crisis management in dementia that could help reduce inappropriate admissions, improve care, and improve quality of service provision. The AQUEDUCT Resource Kit (RK), designed in the AQUEDUCT research programme, includes a Fidelity Measure to determine the areas where the services could improve and a Best Practice Toolkit to provide the teams with resources to develop specific areas of their practice[6].

The aim of this study was to conduct a randomised controlled trial to evaluate the effectiveness of the Resource Kit for TMCDs and to examine staff use of this Resource Kit quantitatively.

## Results

Figure 1 shows the study recruited 23 TMCDs in different geographical locations in England between September 2021 and March 2023; 11 TMCDs were randomised to the intervention arm and 12 to the usual care control arm. Data was missing for one TMCD in the control arm, which did not provide the primary outcome measure at 6 months due to unforeseen changes and errors in their software data management systems. From the 23 TMCDs, 116 TMCD staff were recruited in the intervention arm, while 122 were recruited in the control arm. During the follow-up, 35 people with dementia and carers were recruited to the intervention arm and 40 to the control arm. Information about missing data in our primary and secondary outcome measures can be found in Fig. 1.

The demographic characteristics indicate the distribution of the number of people with dementia at the constituency level was similar between TMCDs in the intervention and control arms The gender distribution among individuals with dementia and their carers revealed a lower proportion of females in the intervention arm ($N = 13$, 48.2%) compared to the control arm ($N = 21$, 67.7%). In contrast, among TMCD staff, the proportion of females was similar across the intervention ($N = 76$, 79.2%) and control arms ($N = 86$, 83.5%). Among TMCD staff, White British individuals predominated, observing 75% ($N = 87$) in the intervention arm and 77% ($N = 94$) in the control arm identified. Similarly, among individuals with dementia and their carers, 74.3% ($N = 26$) in the intervention arm and 72.5% ($N = 29$) in the control arm identified as White British. TMCD staff that chose not to disclose their ethnicity included 19% ($n = 22$) and 15.6% ($n = 19$), whilst 6% ($n = 7$) and 7.4% ($n = 9$) identified as "Other" in intervention and control arms, respectively.

Table 1 shows that the mean number of psychiatric hospital admissions for people with dementia was: 34.0 at baseline and 34.4 at 6 months follow up in the intervention arm; and 28.4 at baseline and 31.4 at 6 months follow up in the control arm. The median number of psychiatric hospital admissions for people with dementia to mental health beds decreased for the intervention arm from 30.0 (IRQ: 11,44) at baseline to 21.0 (IRQ: 3,35) at 6 months, but was unchanged for the

control arm with 26.5 (IRQ: 7.5,43) at baseline and 26.0 (IRQ: 15.5,47) at 6 months follow-up.

For TMCD staff mean GHQ-12 score in both the intervention and control arms was similar at baseline, intervention arm mean 12.4 (SD: 4.7), control arm mean 12.1 (SD: 5.5), and follow up, intervention arm mean 11.2 (SD: 5.0) control arm mean 11.6 (SD: 4.2). The Work Acceptance and Action Questionnaire score was also similar at baseline: intervention arm mean 36.4 (SD: 5.5); mean control arm: 37.0 (SD: 6.5) and 6 months, intervention arm mean 37.0 (SD: 6.5), and control arm mean 36.1 (SD: 6.7). The mean Utrecht Work Engagement Scale decreased slightly for the control arm from 74 at baseline (SD: 11.9) to 71.3 (SD: 12.3) at 6 months and was unchanged for the intervention arm, from 71.7 (SD: 2.7) at baseline and 72.6 (SD: 11.2) at 6 months.

For people with dementia and their carers, the CSQ-8 at 6 months was similar for those in the intervention and control arms: intervention arm median 30 (IQR: 25,32); control arm median 28 (IQR: 24,31). The mean GHQ-12 at 6 months was 16.9 (SD: 7.5) in the control arm and 15.1 (SD: 5.8) in the intervention arm.

### Primary outcome

Table 2a shows the incidence rate ratios (IRR), 95% confidence Interval and $p$-values from our Intention-to-Treat (ITT) analysis for the difference between arms in the number of hospital admissions for people with dementia to mental health beds at 6 months. There were fewer admissions in the control arm than in the intervention arm (IRR: 0.74; 95% CI: 0.37–1.48), but this difference was not statistically significant ($p$-value = 0.397). Psychiatric hospital admissions were higher in the control arm when using baseline psychiatric hospital admissions as offset (IRR: 1.18; 95% CI: 0.78–1.18), but lower when removing outliers with more than 150 psychiatric hospital admissions in 6 months (IRR: 0.96; 95% CI: 0.62–1.48). A similar result was obtained in the complete case (CC) analysis and the three sensitivity analyses in the Supplementary Material (Table S1 and Fig. S1).

### Secondary outcome results for people with dementia and carers

No statistically significant difference was observed between the intervention and the control arms in the General Health Questionnaire score (Coeff: 1.74; 95% CI: −1.42 to 4.90) or the Client Satisfaction Questionnaire score (Coeff: −2.00; 95% CI: −5.01 to 1.01) at 6 months among people with dementia and carers in the intervention and control arms (Table 2b), or in the sensitivity analysis that treated as missing those scores and scales with more than 50% missing items (Table S2 in the Supplementary Material).

### Secondary outcome results for TMCD staff

For TMCD staff, scores on the General Health Questionnaire, the Work Acceptance & Action Questionnaire and the Utrecht Work Engagement Scale, showed no statistically significant difference between the intervention and control arms at 6 months in the Intention-to-Treat analysis (Table 2c). Results were similar in the complete case and sensitivity analysis that set to missing scores and scales with more than 50% missing items (Table S3 in the Supplementary Material).

### Fidelity adherence

Each TMCD was required to implement at least four items from the online Best Practice Toolkit. Meeting these requirements suggested how well the intervention was implemented within the TMCDs. Table 3 shows there was strong fidelity among some TMCDs. Five TMCDs fully adhered to or exceeded fidelity requirements, demonstrating that the Best Practice Toolkit could be successfully implemented when institutional and operational conditions allowed. Of these, two exceeded fidelity goals implementing more tools than required. Four TMCDs faced implementation challenges and only partially adhered, meaning some aspects of the toolkit were not fully integrated into practice. Two TMCDs did not implement any tools.

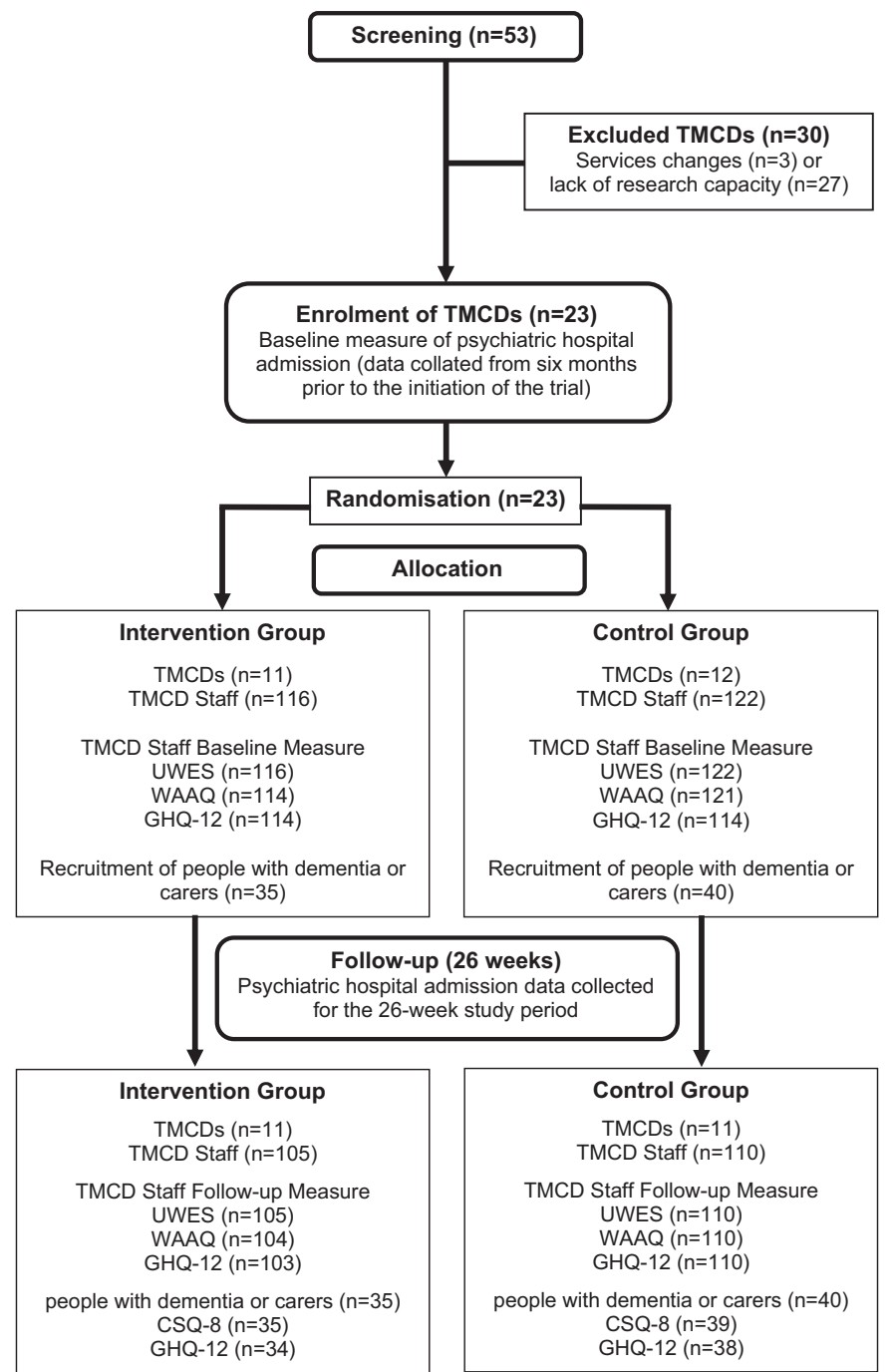

**Fig. 1 | The AQUEDUCT trial consort flow diagram.** Fig. 1, the consort flow diagram illustrates the flow of participants through the multi-site, randomised controlled trial of the online AQUEDUCT Best Practice Tool and Online Resource Kit to support Teams Managing Crisis in Dementia (TMCDs). The Screening panels show the total number of crisis teams assessed for eligibility (*n* = 53), with exclusions noted due to service difficulties or lack of research capability (*n* = 30), resulting in enrolment of 23 TMCDs. The Allocation panels group shows the randomisation of eligible crisis teams into the intervention arm (*n* = 11) or control arm (*n* = 12), plus the recruitment of individual team members into the intervention (*n* = 116) and control arms (*n* = 122). The panels below also show the outcome measures completed, and the recruitment of service users (carers and people with dementia). The Follow up panel shows the withdrawal rate for TCMDs, and the completion rates for the team and service user measures at follow up at 26 weeks.

## Items used from the best practice toolkit

Items used from the Best Practice Toolkit varied across TMCDs, with some items demonstrating higher rates of implementation within TMCDs than others (Table 4). Clinical supervision templates and the overview of service presentation were the most consistently adopted, each implemented by five TMCDs. Quality improvement tools, such as the patient and carer service questionnaires (implemented by four TMCDs), model for monitoring and improvement, and information leaflet template, saw moderate uptake (each implemented by three TMCDs). While operational policy templates were also used by three TMCDs, four others were unable to implement them due to institutional constraints. Notably, discharge-related tools, such as the discharge letter template and daily handover checklist, had the lowest fidelity, implemented by only one TMCD each. Several items, including managerial supervision templates, and clinical supervision templates were considered by two TMCDs but not implemented, and the care

**Table 1 | Descriptive statistics for primary and secondary outcomes by unit of analysis at baseline and six months: TMCD, people with dementia and carers or TMCD staff**

| | Intervention | | Control | |
|---|---|---|---|---|
| | Baseline | 6-month | Baseline | 6-month |
| *Primary outcome* | | | | |
| Number of psychiatric hospital admissions for people with dementia in the past 6 months | | | | |
| Mean (SD) | 34.0 (27.4) | 34.4 (43.7) | 28.4 (22.6) | 31.4 (22.2) |
| Median (Q1,Q3) | 30.0 (11,44) | 21.0 (13,35) | 26.5 (7.5, 43) | 26.0 (15.5, 47) |
| Total TMCD observations | 11 | 10 | 12 | 12 |
| *Secondary outcomes* | | | | |
| Client satisfaction questionnaire score (CSQ-8) | | | | |
| PWD and carers, Median (Q1,Q3) [*n*] | | 30 (25,32) [*n* = 35] | | 28 (24,31) [*n* = 39] |
| General Health Questionnaire score | | | | |
| People with dementia and carers, Mean (SD) [*n*] | | 15.1 (5.8)[*n* = 34] | | 16.86 (7.5) [*n* = 38] |
| TMCD staff, Mean (SD) [*n*] | 12.4 (4.7) [*n* = 114] | 11.2 (5.0) [*n* = 103] | 12.1 (5.5) [*n* = 122] | 11.6 (4.2) [*n* = 110] |
| Work acceptance & action questionnaire score | | | | |
| TMCD staff, Mean (SD) [*n*] | 36.4 (5.5) [*n* = 114] | 36.2 (6.1) [*n* = 104] | 37.0 (6.5) [*n* = 121] | 36.1 (6.7) [*n* = 110] |
| Utrecht work engagement scale | | | | |
| TMCD staff, Mean (SD) [*n*] | 71.7 (12.7) [*n* = 116] | 72.6 (11.2) [*n* = 105] | 74.0 (11.9) [*n* = 122] | 71.3 (12.3) [*n* = 110] |

**Table 2 | Incidence rate ratios (IRR) for primary outcome results and coefficient (Coeff.) for secondary outcome results, 95% confidence interval and *p*-values for the difference between intervention and control arm**

| | *N* | IRR | *p*-value | 95% confidence interval | |
|---|---|---|---|---|---|
| (a) *Primary outcome* | | | | | |
| Psychiatric hospital admissions at 6 months (constituency-level population with dementia as offset) | 23 | 0.74 | 0.397 | 0.37 | 1.48 |
| | *N* | Coeff. | *p*-value | 95% confidence interval | |
| (b) *Secondary outcomes: people with dementia and family carers* | | | | | |
| General Health Questionnaire score (GHQ-12) | 72 | 1.74 | 0.276 | −1.42 | 4.90 |
| Client Satisfaction Questionnaire score (CSQ-8) | 74 | −2.00 | 0.189 | −5.01 | 1.01 |
| (c) *Secondary outcomes: TMCD staff* | | | | | |
| General health questionnaire score (GHQ-12) | 238 | 0.83 | 0.319 | −0.80 | 2.46 |
| Work acceptance & action questionnaire score (WAAQ) | 238 | 0.06 | 0.962 | −2.24 | 2.35 |
| Utrecht work engagement scale (UWES) | 238 | −2.56 | 0.316 | −7.56 | 2.44 |

Intention-to-Treat analysis was computed using multiple imputations in Stata (20 imputations). Missing items in scores and scales were imputed using pro-rating. Quantile regression for CSQ-8 at the median (*P*50 = 29). Missing scores and scale were imputed through multilevel multiple imputation using the jomo R package and calculating *p*-values for TMCD staff secondary outcomes. Post-estimation analyses were performed to check the conversion of parameters generated by Markov chain Montecarlo (MCMC) modelling used in these multilevel multiple imputations.

pathway template was considered, but not implemented by one TMCD. These findings suggest that while some elements of the toolkit were readily adopted, others faced systemic barriers, limiting full fidelity.

## Discussion

The AQUEDUCT programme was designed to improve the management of crises affecting people living with dementia. The comprehensive approach included a scoping review[12], the development of the Best Practice Model and Resource Kit[7], a feasibility study[11], and a qualitative study of crisis management[14], leading to this randomised controlled trial (RCT)[6]. The trial found no difference in the number of psychiatric hospital admissions at six months for people with dementia, and the result was robust to different model specifications and sensitivity analyses. Secondary outcomes for people with dementia and carers, including satisfaction and psychological well being, also did not differ between intervention and control arms. Also, secondary outcomes for TMCD staff, including workplace

psychological flexibility (WAAQ), work engagement (UWES), and mental health (GHQ) symptoms, did not differ between intervention and control arms at follow-up. These results do not provide evidence to support the wider implementation of the AQUEDUCT Resource Kit.

The trial was underpowered, raising concerns regarding the possibility of a Type II error. However, the available data provided no indication that achieving the target of 24 randomised teams would have yielded a different outcome. Furthermore, there was no evidence to suggest that increasing the number of TMCDs in both arms would have been likely to result in a statistically significant difference between the groups in the primary outcome.

Existing literature highlights the challenges associated with recruiting individuals with dementia into research[15], a difficulty that is further exacerbated during periods of crisis and in the aftermath of the COVID-19 pandemic[16]. Despite this, our results indicated a reduction in the median number of psychiatric hospital admissions at 6 months for individuals with dementia in the intervention arm, whereas no change was observed in the control arm. However, the primary analysis, which

incorporated constituency-level data on the number of individuals with dementia as an offset, did not replicate this finding. The limited number of teams included in the analysis may have contributed to this discrepancy.

Fidelity to the Best Practice Toolkit within TMCDs varied significantly, highlighting both facilitators and barriers to implementation. While some teams exceeded or met fidelity requirements, others struggled to implement core components, often due to systemic constraints such as gatekeeping policies and operational limitations. This high variability in implementation suggests inconsistencies in the adoption of best practices across TMCDs. There was also a selective adoption of items within the online Best Practice Toolkit. Certain tools, such as the clinical supervision templates and the service presentation, were widely used (by five TMCDs), indicating perceived utility and ease of implementation. However, other tools, particularly those related to discharge processes and managerial supervision, had much lower uptake, suggesting potential operational or systemic barriers.

Several TMCDs were unable to use key items within the online Best Practice Toolkit, particularly the operational policy template and clinical supervision templates, due to organisational restrictions such as TCMDs being required to use a standard policy across all services. This indicates that while the Best Practice Toolkit provided useful guidance, external constraints prevented full fidelity in some settings. The lack of uniform implementation raises concerns about the effectiveness and consistency of TMCD responses for people with dementia. Whilst some teams were enthusiastic about benefits of the Best Practice Toolkit, others may not have appreciated its potential due to partial or absent implementation.

The trial ran between 2021 and 2023 when there were numerous and enduring challenges related to the Covid-19 global pandemic impacting both on the research process and the NHS services[17,18]. The research, including the training, had to transfer to an exclusively online approach, meaning that contact, training and engagement with the services was limited in scope. As a result of this ongoing accountability and support was very constrained. In contrast, the CORE Study's adult CRHTT service improvement programme, which included structured fidelity reviews, facilitated goal-setting events, continuous external coaching, and a 'learning collaborative practice community', successfully enhanced model fidelity and reduced admissions, underscoring the necessity of comprehensive implementation support for effective resource utilisation[17]. Our study suggests that shows providing tools without continuous training and support fails to produce a 'learning collaborative' community[17] whereby priorities and goals can be achieved and service improvement successes and problem-solving barriers in achieving goals can be shared.

It is important to highlight that the challenges presented to TMCDs by COVID, did not detract from the persistent issues faced by TMCDs, such as under-resourcing, particularly in staff shortages. These ongoing systemic limitations are critical factors influencing the implementation and effectiveness of interventions. In addition, TMCDs were under immense pressure due to high levels of demand, staff sickness, and Covid-19 isolation rules, which varied across time. This led to major delays in the recruitment of TMCDs and the trial. The primary outcome was also impacted as the pandemic patterns of hospital admission varied. Additionally, differences in the specific timing and impact of COVID-19 on each TMCD and the related home based and other community services, consequently undermined the trial and intervention processes in varying ways.

Additionally, the outcome measure of hospital admissions could have been further standardised by controlling for the number of psychiatric beds for people with dementia in each area, since higher and lower proportions of beds can shape thresholds of admission and, therefore, the capacity of the intervention to shape crisis management. Moreover, it was very difficult to get data on people with dementia and carers during follow-up which resulted in small numbers for the

**Table 3 | Fidelity adherence**

| Fidelity adherence | TMCD count |
|---|---|
| Exceeded fidelity requirements | 2 |
| Met fidelity requirements | 3 |
| Partially met fidelity requirements | 4 |
| Failed to meet fidelity requirements | 2 |

**Table 4 | Items used from the online best practice toolkit**

| Best practice toolkit item | Count of TMCDs | Implemented |
|---|---|---|
| Clinical supervision template (1) | 5 | Yes |
| Clinical supervision template (2) | 5 | Yes |
| Overview of service presentation | 5 | Yes |
| Patient and carer service questionnaires | 4 | Yes |
| Model for Monitoring and Improvement | 3 | Yes |
| Operational policy template | 3 | Yes |
| Information leaflet template | 3 | Yes |
| Referral audit template | 2 | Yes |
| Daily handover checklist | 2 | Yes |
| Template for discharge letter | 1 | Yes |
| Discharge letter template | 1 | Yes |
| Operational policy template | 4 | No |
| Managerial supervision template | 2 | No |
| Clinical supervision template (1) | 2 | No |
| Clinical supervision template (2) | 2 | No |
| Care pathway template | 1 | No |

analysis of secondary outcomes. Data collection was undertaken remotely, which may have increased the probability of low response rates from people with dementia and their carers especially in the period following a crisis when then may have too stressed or exhausted to consider completing questionnaires.

Further work should investigate the causes of crisis so that approaches can be developed for community services to help prevent crisis developing and deteriorating. However, there have been few studies of dementia crisis services as highlighted in a recent review[12], and other models such as a specialist crisis intervention programme have also shown promising results including reduced nursing home admissions[19]. However, trying to reorientate services to a standard model made this a highly complex trial, and even without the Covid-19 pandemic. Service change can be much more challenging compared to a simple easily defined intervention. In particular, trying to change the outcomes of crisis in people with dementia is especially challenging given the added complications of physical health needs, frailty, ageing, and stress applying to both people with dementia and their carers, who may also be at risk of general hospital admissions due to a wide range of physical health needs or accidents. The complex interplay of medical, social, and environmental determinants influencing crisis situations in dementia care requires multifaceted interventions tailored to individualised needs, which may in any case have changed since the COVID-19 pandemic[20,21].

Hence, the TCMDs could be an important factor, but one of many other factors influencing psychiatric hospital admissions[4,9]. Equally, TMCDs, faced with pressures from staff shortages, lack of resources[4,19], and burn-out[19], may have lacked the capacity to fully implement the RK materials. In contrast, during the trial, staff from TMCDs were enthusiastic about the value of the RK in facilitating quality improvement initiatives, standardising procedures and providing a framework for better practice. However, feedback also highlighted difficulties in

implementing the RK because of their obligations to use agreed local processes/policies determined by their own NHS Trust or the challenging nature of navigating organisational hierarchies which posed difficulties for frontline staff. Also the dissemination of information about AQUEDUCT within both arms could have an educational and awareness-raising effect; hence, exposing TMCD staff to information about the need for evidence-based practices and guidelines. This could prompt both intervention and control arms to try and improve practice. However, the pattern of results did not suggest such an improvement in either group, indicating that this was unlikely.

The high GHQ scores in carers and people with dementia were suggestive of mental health issues, the stress of crises[20], and may reflect exposure to the distress and cognitive decline of people with dementia leading to possible compassion fatigue among carers[21,22]. Also TMCD staff in both arms had high GHQ-12 scores suggesting psychological distress[22], which is in line with other studies[21–24] and may illustrate the unprecedented challenges mental health crisis services faced during the COVID-19 pandemic, including staff shortages and burnout, limited access to referral services, and anxiety about infection risks[17,18]. Individuals who choose to work as TMCD staff in dementia care often possess strong empathic tendencies, but this could also make them vulnerable to emotional distress[16,17]. Additionally, empathy and awareness may also promote higher self-reported rates of psychological ill-being than the general population. Personal coping strategies and resilience factors play a role in determining how staff members manage these stressors and their overall mental health[15–17].

Given the high GHQ-12 scores indicating significant staff distress, future interventions could incorporate targeted supports such as peer networks and stress management training. These measures could enhance staff well-being and improve intervention fidelity and effectiveness by enabling increased staff capacity to fully engage with the intervention, improve focus, adherence to protocol and optimise delivery and impact of the intervention.

Staff had average to high UWES scores, indicating good work engagement, suggesting staff are emotionally invested in their work, finding it meaningful and deriving satisfaction from helping others. This is consistent with NHS England's 2024 national NHS survey that showed high levels of work engagement (72.8%) alongside high levels of burnout (42.7%) with 34.2% of staff finding work emotionally exhausting[18]. Our results suggest TMCD practitioners struggle with their mental health and wellbeing despite high levels of engagement and commitment with their role. Staff may have used work engagement as a coping mechanism to mitigate against poor mental health, increasing the likelihood of burnout[15,18].

The AQUEDUCT RCT did not show a decrease in the number of psychiatric hospital admissions after six months of implementation of the RK. The finding of lack of significant benefits following implementation of the Resource Kit should be viewed with caution because of the limitations of the study, including the unprecedented demands on health services during the COVID-19 pandemic. Hence, a case could be made for further evaluation under more stable conditions and when additional mechanisms to get the Resource Kit into practice have been developed.

Secondary outcomes reveal that, despite high work engagement, TMCD staff experienced poor mental health in both the intervention and control groups. These findings suggest that working in TMCDs is highly demanding and stressful, affecting both the implementation of the RK and the mental well-being of practitioners, despite their commitment and high engagement levels. Staff showed enthusiasm for the Resource Kit and a strong desire to enhance TMCD services were concerned about the time needed and the many practical barriers to quality improvement. Collectively, these observations underscore the importance of considering contextual factors, organisational dynamics, and stakeholder engagement in interpreting study outcomes and

informing future interventions aimed at enhancing crisis management practices in dementia care contexts.

While the Best Practice Toolkit has the potential to be useful for dementia crisis care, several teams struggled to implement it effectively due to policy constraints and variability in organisational readiness. Future work should consider embedding structured implementation support, fostering a learning collaborative, and addressing external constraints to enhance adoption and sustainability of best practices in dementia crisis care. The experience of navigating these challenges can inform the research processes needed for other complex studies and provide a model for developing guidelines for research in challenging service settings.

## Methods
### Ethics approval and regulatory compliance
This study was conducted in accordance with all relevant ethical guidelines and regulations. As such, all participants were provided with written information about the study and given adequate time to ask questions and consider participation. Consent was voluntary and documented, and participants were informed of their right to withdraw at any time without consequence. Ethical approval for the AQUEDUCT trial was granted by the Health Research Authority for England and Wales, (REC reference: 14/EM/0233), sponsored by Nottinghamshire Healthcare NHS Foundation Trust, as per study protocol[6].

### Study design and setting
This was a pragmatic two-arm, randomised, cluster, parallel-group, treatment-as-usual (TAU) controlled trial[6]. The full protocol and statistical analysis plan is available via open access publication[6]. Treatment allocation was a 1:1 ratio, and TMCDs were randomised to either the Resource Kit (RK) plus usual care intervention arm or usual care control arm. The null hypothesis was that there was no difference in the effect of crisis care management between TMCDs using the RK and those offering usual care.

The primary objective of AQUEDUCT trial was to evaluate the impact of the use of RK on psychiatric hospital admissions (primary outcome) for people with dementia in the geographical catchment area covered by the TMCD[6]. Secondary objectives included evaluating the impact of TMCDs using the RK on acute/general hospital admissions for people with dementia in the geographical catchment area covered by the TMCD; on service use for both people with dementia and carers; and on TMCD staff. Both primary and secondary outcomes tested for the superiority of the use of the RK by TMCDs over TAU.

The trial was conducted within NHS Trusts across England that operated a TMCD service and could supply regional psychiatric hospital admission data for people with dementia and understood and agreed to the randomisation allocation process. Recruitment of participants was limited; to staff working in participating TMCDs, and to people with dementia or carers who had accessed the services. The first TCMD was enroled on 22.10.2021 and the last TCMD was enroled on 18.01.2023.

### Participants
Participants included the TMCDs, staff members of the TMCDs, and people with dementia and carers supported by these teams. Given that TMCDs provide integrated care for people with dementia and information and training to family carers, both groups were analysed together as part of the intervention. For TMCDs to be eligible, they had to manage mental health crises for people with dementia living in the community, which included providing home-based mental health assessments and interventions. TMCDs were excluded if they shared immediate management, administrative or core clinical staff, or the same office with another team already randomised in the study. They

were also excluded if they were undergoing or expecting to undergo a major service reorganisation during the study period, or if a team leader who was previously exposed to the intervention subsequently became the lead for a potential team to be randomised in the Trial.

## Intervention

The RK intervention was an online resource for TMCDs with two components. The Fidelity Measure enabled TMCDs to evaluate their practice according to 50 Best Practice Statements in relation to the crisis service, rapid crisis assessment and intervention, and service resources. Practice was evaluated using evidence provided by staff, a review checklist for people with dementia and their carers in relation to the service definition of crisis, and evidence of service quality improvement was discussed in business meetings. The Best Practice Toolkit, included resources for teams to improve and develop their practice, including operational policy templates, an overview of service presentations, referral and supervision templates, satisfaction questionnaires, compliment slip templates, daily handover templates, and a managerial supervision template. The RK was available as a password-protected online resource. The research team provided initial training (2–3 h) on using the Best Practice Toolkit once the Fidelity Measure was initially completed. More details can be found in the Study Protocol[6] and Trial registration: ISRCTN 42855694; Registered on 04/03/2021; Protocol number: 127686/2020v9. Research Ethics Committee, 09/03/2021, Ref. 15/WM/0004; IRAS ID: 289982.

## Procedures

Recruitment was purposive and sought to reflect the diversity of team service models and service user demographics, as found in a previous study[12]. NHS Trust sites were recruited via professional and research networks across England, and Trusts identified teams for inclusion in the study. Trusts had to confirm their capacity to undertake the Trial by completing the Health Research Authority Statement of Activities, which constituted a formal agreement with the study sponsor. In each TMCD, the research team briefed an Individual Team Manager or delegated senior practitioner about the Trial and provided participant information sheets. Two staff members on each team acted as volunteer research coordinators, who were responsible for recruiting the remaining staff from the team.

Involvement in the Trial included collecting hospital admission data from all NHS Trusts at baseline and 6-month follow-up; initial set-up; completion of the Fidelity Measure before and after implementation of the RK; and delivery of all participant-completed measures. TMCDs in the intervention arm implemented the RK for 6 months in addition to usual care, and TMCDs in the control arm delivered usual care only for 6 months.

During the follow-up period, people with dementia and carers were identified from referrals to the team's caseload. They were approached by TCMD staff, who explained the team's participation in the trial and invited them to participate. If agreeable they were provided with an information sheet, an opportunity to ask questions, and up to three days to decide if they wished to participate. Consent was obtained from the person with dementia and carers separately.

Consent from the two TMCD research coordinators was obtained during the site set-up visit in each TMCD. They also arranged and confirmed consent from their team colleagues, following the same procedure used to confirm their own consent. All participants were informed of their right to withdraw from the research for any reason and at any time and that this decision would not impact their current or future work within clinical services or access to and use of services.

## Study arms

TMCDs in the intervention arm completed the Fidelity Measure before the intervention phase to determine areas where practice could be improved. During the 6-month implementation phase, TMCDs were asked to implement at least four relevant templates out of the 25 available from the Best Practice Toolkit. The Fidelity Measure was repeated at the end of the 6-month intervention phase[6].

TMCDs in the control arm did not have access to the RK, did not complete the Fidelity Measure, and did not use elements of the Best Practice Toolkit during the 6-month implementation phase of the trial but continued with their usual practice. Both arms were free to undertake all usual care interventions.

## Randomisation and blinding

Once consent was obtained, each TMCD was entered into a remote web-based randomisation system and randomly assigned to one of two arms, either intervention arm (using the Resource Kit) or usual care, with an equal chance of allocation to either group. Allocation was determined by a computer-generated pseudo-random code using random permuted blocks of varying size, stratified by the population size (number of people with dementia) in each TMCD catchment area. People with dementia, carers, outcome assessors and statisticians were blinded to TMCD arm allocation until the data analysis was completed. Blinding of TMCDs to receipt of the RK was not possible. AQUEDUCT Team members involved in data analysis remained blind throughout. Details of the randomisation method are held securely within the statistics master file.

## Data collection

Data were collected from NHS Trusts, TMCDs, staff, and new referrals of people with dementia and carers managed by these TMCDs. Psychiatric hospital admissions data in the TMCD catchment area (primary outcome) for the preceding 6 months were collated and reported retrospectively by the relevant NHS Trust Department at baseline and six-month follow-up. Similarly, the research team retrospectively collected acute/general hospital admissions data for people with dementia from each TMCD catchment area. Access to anonymized primary datasets (generated during the study) will be available via the first author.

## Outcomes

The primary outcome was the number of hospital admissions for people with dementia to mental health beds in the geographical catchment area of the TMCD (as defined by postcode), which was reported at baseline and the primary endpoint of 6-month follow-up. Secondary outcomes included hospital admissions for people with dementia to acute beds in the geographical catchment area of the TMCD (as defined by postcode). No data was obtained on hospital admissions for people with dementia in acute general hospital beds.

For TMCD staff, there were three outcomes measured at baseline and six months: the 7-item Work Acceptance and Action Questionnaire, or the WAAQ score, measuring psychological flexibility in the workplace through seven items related to the ability and willingness to continue engaged with work while experiencing distress and other emotions[23]; the 17-item Utrecht Work Engagement Scale or UWES scale, measuring vigour, dedication and absorption at work[24]; and the 12-item General Health Questionnaire or GHQ-12 score, measuring mental health and psychological wellbeing[25].

For people with dementia and carers, there were two outcomes measured only at six months: service satisfaction, measured through the Client Satisfaction Questionnaire (CSQ-8) score[26], and mental wellbeing using the General Health Questionnaire (GHQ-12) score[25].

Lastly, the Fidelity Measure measured practice quality in managing crises was completed by TMCDs in the intervention arm only at baseline and 6 months[6].

## Sample size

The sample size calculation was based on scoping information collected in earlier stages of the AQUEDUCT research programme, which

showed an average of 33 hospital admissions per TMCD catchment area over a period of 6 months. Following consultation with stakeholders, it was agreed that a 20% reduction represented the minimum clinically important difference (MCID). We required 15 TMCDs in each of the two study arms (30 in total) to detect a 7-point mean admission count difference between arms with 90% power at a two-tailed 0.05 significance level[27,28], assuming the count of hospital admissions follows a Poisson distribution. Following trial commencement, a non-substantial amendment was made reducing the target sample size of 30 TMCDs (per arm) to 24 (12 per arm) due to delays and difficulties with recruitment and service pressures observed associated with the COVID-19 pandemic. The decision was made by the Trial Management Group in consultation with the funder, resulting in a corresponding reduction in statistical power from 90% to 80%. Recruitment in the time available was only able to achieve 23 TCMDs not the planned 24.

## Statistical analysis

The treatment effect estimate on the primary outcome was quantified by incidence rate ratio (IRR) by means of negative binomial regression modelling, informed by data exploratory. The binary treatment arm variable was used as an explanatory variable to quantify the treatment effect estimates on psychiatric hospital admissions at six months for people with dementia to mental health beds in the geographical catchment area of the TMCD. The offset variable was the constituency-level number of people with dementia. All analysis was conducted on an intention-to-treat (ITT) basis and used multiple imputations based on the analytical model, assuming data at 6 months was missing at random (MAR). We were unable to formally test this assumption for the primary outcome due to only one missing value and a small overall sample size. Given this limitation, we proceeded under the MAR assumption. However, complete-case (CC) results were presented in the Supplementary Material and closely aligned with the ITT results, suggesting that missing data did not considerably influence our findings. We performed three sensitivity analyses for the primary outcome using the same negative binomial model but with three variations. The first used psychiatric hospital admissions for people with dementia to mental health beds in the geographical catchment area of the TMCD at baseline as the offset. The second used the same offset as the analytical model but excluded the observation with an extreme value in the primary outcome (more than 150 psychiatric hospital admissions in 6 months). The third used the same offset as in the analytical model but used psychiatric hospital admissions at baseline as a covariate.

All secondary outcome measures used linear regression after performing distributional checks, and we used pro-rating to impute missing items in scores and scales. For secondary outcome measures related to people with dementia and their carers, we performed two sensitivity analyses: (a) setting as missing scores and scales with more than 50% missing items and (b) adjusting by gender (Table S3 in the Supplementary Material). For secondary outcome measures related to TMCD staff, we performed ITT analysis using multilevel multiple imputations, considering TMCD staff nested in TMCDs, where missing scores and scales were imputed using the jomo R package for multilevel multiple imputation[8]. We assumed data were Missing At Random (MAR) and performed Little's chi-squared test to assess the Missing Completely At Random (MCAR) assumption. We performed two sensitivity analyses: (a) complete-case results and (b) multiple imputation results adjusted by gender, which can be found in Table S2 in the Supplementary Material.

All analyses were performed in Stata 18 and R, and results were reported using 95% confidence intervals and p-values. Only analyses prespecified in the protocol were conducted. No measure of acute hospital admissions for people with dementia to acute beds could be collected as the hospitals could not supply the data within the required time frame.

## Reporting summary

Further information on research design is available in the Nature Portfolio Reporting Summary linked to this article.

## Data availability

The anonymised data generated in this study and the REDCap code book have been deposited in the Figshare database under the accession code 29155166 The processed data, including summary tables are provided in Supplementary Material.

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

## Acknowledgements

This study was funded by the National Institute for Health Research (NIHR RP-PG-0612-20004). We thank all the staff involved from NHS Trusts across England, particularly the Teams Managing Crisis in Dementia (TMCD) services that participated in the AQUEDUCT research programme. We extend our gratitude to the people with dementia and their carers who generously contributed their time and insights. We also thank Christine Bailey for her hard work and dedication to the AQUE-DUCT programme, as well as David Prothero our PPI lead and other Patient and Public involvement (PPI) members for their invaluable input and contributions.

## Author contributions

Martin Orrell: Conceptualisation, funding acquisition, investigation, methodology, project administration, resources, supervision, validation, writing original draft, editing and reviewing. Linda O'Raw: Writing original draft, data curation, methodology, project administration, supervision, editing and reviewing. Donna Maria Coleston: Investigation, methodology, project administration, supervision. Magdalena Opazo Breton: Writing original draft, data curation, formal analysis, visualisation, editing and reviewing. Boliang Guo: Data curation, formal analysis, supervision, validation, editing, and reviewing. Tom Dening: Conceptualisation, funding acquisition, investigation, editing, and reviewing. Juanita Hoe: Conceptualisation, funding acquisition, investigation, editing, and reviewing. Brynmor Lloyd-Evans: Conceptualisation, funding acquisition, investigation, editing, and reviewing. Esme Moniz-Cook: Conceptualisation, funding acquisition, investigation, editing, and reviewing. Fiona Poland: Conceptualisation, formal analysis, funding acquisition, investigation, methodology, supervision, validation, editing, and reviewing. Marcus Redley: Data curation, formal analysis, editing, and reviewing. Angela Worden: Editing and reviewing. David Challis: Conceptualisation, funding acquisition, investigation, methodology, supervision, editing, and reviewing.

## Competing interests

The authors declare no competing interests.

## Additional information

M. Orrell ®¹ ✉, L. O'Raw ®¹, D. M. Coleston¹, M. Opazo Breton ®¹, B. Guo¹, T. Dening ®¹, J. Hoe ®², B. Lloyd-Evans³, E. Moniz-Cook⁴, F. Poland⁵, M. Redley⁵, A. Worden¹ & D. Challis ®¹

¹Institute of Mental Health, University of Nottingham, Nottingham, UK. ²Geller Institute of Ageing and Memory, University of West London, London, UK. ³Division of Psychiatry, University College London, London, UK. ⁴Faculty of Health Sciences, University of Hull, Hull, UK. ⁵School of Health Sciences, University of East Anglia, Norwich, UK. ✉e-mail: M.Orrell@nottingham.ac.uk

