## [Transparent Peer Review file · Nature Communications]

Achieving Quality and Effectiveness in Dementia Using Crisis Teams (AQUEDUCT): A randomised controlled trial evaluating the impact of a Best Practice Resource Kit used by Teams Managing Crisis in Dementia

Corresponding Author: Professor Martin Orrell

Version 0:

Reviewer comments:

Reviewer #1

(Remarks to the Author)

The authors have made a thoughtful effort to address the reviewers' comments, improving clarity and contextualization of findings. Below are suggestions for your consideration:

- While the authors clarified missing data sources and used multiple imputation, they did not provide evidence (e.g., Little's test) to support the "missing at random" assumption. This weakens confidence in the imputation's validity.
- The gender imbalance between groups was acknowledged but not statistically controlled for in analyses. Adjusting for baseline imbalances (e.g., via regression models) would strengthen causal inferences.
- The authors defended the intervention's flexibility but did not measure or report fidelity (e.g., how teams used the Resource Kit). However, the lack of data on RK implementation fidelity undermines interpretation of null results. Without knowing how teams used the RK, it is unclear whether the intervention itself was ineffective or poorly executed.
- Provide full protocol (e.g., RK components, training materials) may be helpful for reproducibility.
- The trial was underpowered (23 TMCDs vs. target 24; 75 people with dementia/carers). This raises concerns about Type II error.

Version 1:

Reviewer comments:

Reviewer #1

(Remarks to the Author)

The authors have invested substantial effort in addressing the reviewers' comments, substantially enhancing the manuscript's rigor and transparency. I have no further comments and endorse this paper for publication in Nature Communications.

REVIEWER COMMENTS

AUTHORS COMMENTS

We sincerely thank the reviewer for their comments and hope they feel the changes made are reflective and satisfactory.

Reviewer #1 (Remarks to the Author):

The authors have made a thoughtfully effort to address the reviewers' comments, improving clarity and contextualization of findings. Below are suggestions for your consideration:

1) While the authors clarified missing data sources and used multiple imputation, they did not provide evidence (e.g., Little's test) to support the "missing at random" assumption. This weakens confidence in the imputation's validity.

Author's response:

We acknowledge the reviewer's point regarding testing the Missing At Random (MAR) assumption. In our study, the primary outcome had only one missing value, and the overall sample size was small, limiting our ability to conduct formal tests such as Little's MCAR test due to insufficient degrees of freedom. Despite this limitation, the results from our complete-case sensitivity analysis were consistent with those from the multiple imputation approach, suggesting that the missing data had minimal impact on our findings. We have clarified this point in the Statistical Analysis section.

"The treatment effect estimate on the primary outcome was quantified by Incidence Rate Ratio (IRR) by means of negative binomial regression modelling, informed by data exploratory. The binary treatment arm variable was used as an explanatory variable to quantify the treatment effect estimates on psychiatric hospital admissions at six months for people with dementia to mental health beds in the geographical catchment area of the TMCD. The offset variable was the constituency-level number of people with dementia. All analysis was conducted on an intention-to-treat (ITT) basis and used multiple imputations based on the analytical model, assuming data at

six months was missing at random (MAR). We were unable to formally test this assumption for the primary outcome due to only one missing value and a small overall sample size. Given this limitation, we proceeded under the MAR assumption. However, complete-case (CC) results were presented in the Supplementary Material and closely aligned with the ITT results, suggesting that missing data did not considerably influence our findings. We performed three sensitivity analyses for the primary outcome using the same negative binomial model but with three variations. The first used psychiatric hospital admissions for people with dementia to mental health beds in the geographical catchment area of the TMCD at baseline as the offset. The second used the same offset as the analytical model but excluded the observation with an extreme value in the primary outcome (more than 150 psychiatric hospital admissions in six months). The third used the same offset as in the analytical model but used psychiatric hospital admissions at baseline as a covariate.”

This showed Little Test is still valid to test MAR assumption. It is well accepted all assumption on the missingness mechanism was untestable.

(Please view **pages 7 and 8** of the manuscript).

For the secondary outcome variables using multiple imputations, we were able to perform Little’s MCAR test, and the results indicated that the MCAR assumption did not hold. This has now been clarified in the Statistical Analysis section.

“All secondary outcome measures used linear regression after performing distributional checks, and we used pro-rating to impute missing items in scores and scales. For secondary outcome measures related to people with dementia and their carers, we performed two sensitivity analyses: a) setting as missing scores and scales with more than 50% missing items, and b) adjusting by gender (Table S3 in the Supplementary Material). For secondary outcome measures related to TMCD staff, we performed ITT analysis using multilevel multiple imputations, considering TMCD staff nested in TMCDs, where missing scores and scales were imputed using the jomo R package for multilevel multiple imputation [8]. We assumed data were

Missing At Random (MAR) and performed Little's chi-squared test to assess the Missing Completely At Random (MCAR) assumption. The test results indicated that the MCAR assumption did not hold. We performed two sensitivity analyses: a) complete-case results, and b) multiple imputation results adjusted by gender, which can be found in Table S2 in the Supplementary Material."

(Please view page 9 of the manuscript)

2) The gender imbalance between groups was acknowledged but not statistically controlled for in analyses. Adjusting for baseline imbalances (e.g., via regression models) would strengthen causal inferences.

Author's response:

"We have now included an additional sensitivity analysis that adjusts by gender our secondary outcome results for TMCD staff and for people with dementia and their family carers. Results can be found in Table S2 and Table S3 in the Supplementary Material. We added the following in the Statistical Analysis section:

"All secondary outcome measures used linear regression after performing distributional checks, and we used pro-rating to impute missing items in scores and scales. For secondary outcome measures related to people with dementia and their carers, we performed two sensitivity analyses: a) setting as missing scores and scales with more than 50% missing items, and b) adjusting by gender (Table S3 in the Supplementary Material). For secondary outcome measures related to TMCD staff, we performed ITT analysis using multilevel multiple imputations, considering TMCD staff nested in TMCDs, where missing scores and scales were imputed using the jomo R package for multilevel multiple imputation [8]. We assumed data were Missing At Random (MAR) and performed Little's chi-squared test to assess the Missing Completely At Random (MCAR) assumption. We performed two sensitivity analyses: a) complete-case results, and b) multiple imputation results adjusted by gender, which can be found in Table S2 in the Supplementary Material."

(Please view page 9 of the manuscript)

3) The authors defended the intervention's flexibility but did not measure or report

fidelity (e.g., how teams used the Resource Kit). However, the lack of data on RK implementation fidelity undermines interpretation of null results. Without knowing how teams used the RK, it is unclear whether the intervention itself was ineffective or poorly executed.

Author's response:

We have added more detail to demonstrate that fidelity was assessed and is now reported:

“Fidelity Adherence

Each TMCD was required to implement at least four items from the online Best Practice Toolkit. Meeting these requirements suggested how well the intervention was implemented within the TMCDs. Table 3 shows there was strong fidelity among some TMCDs. Six TMCDs fully adhered to or exceeded fidelity requirements, demonstrating that the Best Practice Toolkit could be successfully implemented when institutional and operational conditions allowed. Of these, two exceeded fidelity goals implementing more tools than required. Four TMCDs faced implementation challenges and only partially adhered, meaning some aspects of the toolkit were not fully integrated into practice. Two TMCDs did not implement any tools.”

(Please view page 12 manuscript)

And:

“Items Used from the Best Practice Toolkit

Items used from the Best Practice Toolkit varied across TMCDs, with some items demonstrating higher rates of implementation within TMCDs than others (Table 4). Clinical supervision templates and the overview of service presentation were the most consistently adopted, each implemented by five TMCDs. Quality improvement tools, such as the patient and carer service questionnaires (implemented by four TMCDs), model for monitoring and improvement, and information leaflet template, saw moderate uptake (each implemented by three TMCDs). While operational policy templates were also used by three TMCDs, four others were unable to implement them due to institutional constraints. Notably, discharge-related tools, such as the discharge letter template and daily handover checklist, had the lowest fidelity,

implemented by only one TMCD each. Several items, including managerial supervision templates, and clinical supervision templates were considered by two TMCDs but not implemented, and the care pathway template was considered, but not implemented by one TMCD. These findings suggest that while some elements of the toolkit were readily adopted, others faced systemic barriers, limiting full fidelity.” (Please view **pages 12 and 13** in the manuscript)

In addition we have added the following to the Discussion section of the manuscript:

“Fidelity to the Best Practice Toolkit within TMCDs varied significantly, highlighting both facilitators and barriers to implementation. While some teams exceeded or met fidelity requirements, others struggled to implement core components, often due to systemic constraints such as gatekeeping policies and operational limitations. This high variability in implementation suggests inconsistencies in the adoption of best practices across TMCDs. There was also a selective adoption of items within the online Best Practice Toolkit. Certain tools, such as the clinical supervision templates and the service presentation, were widely used (by five TMCDs), indicating perceived utility and ease of implementation. However, other tools, particularly those related to discharge processes and managerial supervision, had much lower uptake, suggesting potential operational or systemic barriers.

Several TMCDs were unable to use key items within the online Best Practice Toolkit, particularly the operational policy template and clinical supervision templates, due to organisational restrictions such as TCMDs being required to use a standard policy across all services. This indicates that while the Best Practice Toolkit provided useful guidance, external constraints prevented full fidelity in some settings. The lack of uniform implementation raises concerns about the effectiveness and consistency of TMCD responses for people with dementia. Whilst some teams were enthusiastic about benefits of the Best Practice Toolkit, others may not have appreciated its potential due to partial or absent implementation.”

(Please see **pages 14 and 15** in the manuscript)

We have also made the following changes to the conclusion:

“While the Best Practice Toolkit has the potential to be useful for dementia crisis care, several teams struggled to implement it effectively due to policy constraints and variability in organisational readiness. Future work should consider embedding structured implementation support, fostering a learning collaborative, and addressing external constraints to enhance adoption and sustainability of best practices in dementia crisis care. The experience of navigating these challenges can inform the research processes needed for other complex studies and provide a model for developing guidelines for research in challenging service settings.”

(Please view **pages 18 and 19** of the manuscript)

Tables 3 and 4 have been added. Please view **page 25** of the manuscript.

4) Provide full protocol (e.g., RK components, training materials) may be helpful for reproducibility.

A protocol for the main trial is provided within the manuscript [6] (please view page 4). However, this has been reiterated at the start of the methods section, and made clearer in the manuscript.

Please view **page 2** of the manuscript.

5) The trial was underpowered (23 TMCDs vs. target 24; 75 people with dementia/carers). This raises concerns about Type II error.

Author’s response:

Added to the discussion

“The trial was underpowered, raising concerns regarding the possibility of a Type II error. However, the available data provided no indication that achieving the target of 24 randomised teams would have yielded a different outcome. Furthermore, there was no evidence to suggest that increasing the number of TMCDs in both arms

would have been likely to result in a statistically significant difference between the groups in the primary outcome.”

(Please see pages 13 in the manuscript)

And:

“Existing literature highlights the challenges associated with recruiting individuals with dementia into research [21], a difficulty that is further exacerbated during periods of crisis and in the aftermath of the COVID-19 pandemic [22]. Despite this, our results indicated a reduction in the median number of psychiatric hospital admissions at six months for individuals with dementia in the intervention arm, whereas no change was observed in the control arm. However, the primary analysis, which incorporated constituency-level data on the number of individuals with dementia as an offset, did not replicate this finding. The limited number of teams included in the analysis may have contributed to this discrepancy.”

(Please see pages 14 in the manuscript)